Complexity welcome: Pangenome graphs for
comprehensive population genomics.
*Quantitative Plant Biology*, **6:**e43, 1–11

complex variation; pangenome graphs; plant
genomics.

**Corresponding author:**
Detlef Weigel;
Email: weigel@tue.mpg.de

**Associate Editor:**
Dr. Luke Dunning

# Complexity welcome: Pangenome graphs for comprehensive population genomics

Zhigui Bao[1] and Detlef Weigel[1,2] (iD)

[1]Department of Molecular Biology, Max Planck Institute for Biology Tübingen, 72076 Tübingen, Germany; [2]Institute for
Bioinformatics and Medical Informatics, University of Tübingen, 72076 Tübingen, Germany

## Abstract

Pangenome graphs are revolutionising evolutionary and population genomics by moving
beyond linear reference genomes to represent the full spectrum of sequence diversity within
and across species. This review traces the field's progression from reference-augmented graphs
to assembly-based, alignment-first approaches that capture complex structural variation with
reduced bias. We examine key strategies for graph construction, genotyping and implement-
ing graph-aware tools in functional genomics, including transcriptomics and epigenomics.
While much of the work to date has focused on humans, diverse and structurally complex
plant genomes pose unique challenges that require further methodological innovation. Key
bottlenecks – including visualisation, scalability and integration with multi-omic data – persist.
By outlining trade-offs among current tools and emphasising the need for rigorous evaluation
frameworks, we argue that progress will depend on community-driven efforts to unify graph
construction, genotyping and interpretation. Despite technical hurdles, pangenome graphs
offer a powerful foundation for more inclusive evolutionary and population genomics.

## 1. Introduction

The first papers describing nearly complete genome sequences were typically entitled 'The
genome of ...'. Implicit in these titles was the assumption that much of the genome is shared
between different members of a species. Looking back today, it is clear that a great deal has
been learned from the study of genes conserved in each species, the proteins they encode, their
regulatory elements and so on. Similarly, the commonalities as well as fixed differences between
both closely and more distantly related species have greatly informed biology, including our
understanding of evolution over different time scales.

As interesting as all this knowledge is, we cannot fully understand biology without
considering the genetic differences between individuals. These are not only at the root of
adaptation to specific environments but also underlie susceptibility to disease and abiotic factors.
Such seemingly deleterious variants are often linked by evolutionary trade-offs, where genes and
alleles that are favourable in one environment become a liability in a different environment.
The original genome papers for different species, therefore, were often quickly followed by
attempts to record interindividual differences at the whole-genome level. Pioneering in this
regard were scientists working with the plant *Arabidopsis thaliana*. The initial genome paper
already contained information on shotgun sequences from a second strain in addition to the
one from which the genome sequence had been primarily generated (The Arabidopsis Genome
Initiative, 2000). Soon thereafter, the first genome-wide polymorphism analyses were published,
at increasingly higher resolution, relying on a series of different technologies (Clark et al., 2007;
Kim et al., 2007; Nordborg et al., 2002, 2005). Some of the approaches could interrogate in
principle every position in the reference genome, but were limited by the extent of sequence
divergence that could be recorded, and in highly divergent or missing regions, the exact sequence
remained unknown (Clark et al., 2007). This remained the case when much cheaper short-
read sequencing entered the scene, although with increasing lengths of short reads, more and
more of the genome became accessible to polymorphism analyses (Ossowski et al., 2008)].
Importantly, short-read sequencing provided access to sequences that were not present in the

original reference genome, and some of these could even be anchored to reference positions (Cao et al., 2011; Gan et al., 2011; Long et al., 2013; Ossowski et al., 2008; Schneeberger et al., 2011). The picture for other plant species, especially the crops rice and maize, was broadly similar, albeit typically with a delay of a few years (Gao et al., 2019; Li et al., 2014; Zhao et al., 2018).

The use of short reads for the identification of sequence polymorphisms that distinguish the focal individual from the reference strain begins with the mapping of reads against the reference genome sequence. A limitation, therefore, is the reference bias that is caused by the degree of mismatch between a specific short read and its target. Although mismatches may still allow confident mapping, more reads can be mapped with a more closely related genome. This insight led early on to the suggestion of producing synthetic reference genome sequences that represented all possible combinations of polymorphisms across the genome, including those not yet discovered in the genomes analysed (Schneeberger et al., 2009).

A less obvious problem has been that a specific sequence might be present only once in one genome but multiple times in another genome. And even if a sequence is present only once, it might occur in different regions of the genome across individuals. In both cases, short-read mapping can be misleading. For example, heterozygosity may be inferred when in reality there are two closely related but not identical duplicated fragments in the short-read sequenced genome.

Rather than relying on a single reference genome, the concept of the pangenome was introduced to capture all sequence variation within a species. First applied in bacterial genomes (Tettelin et al., 2005), this framework quantified core and dispensable genes across populations, highlighting extensive diversity. As sequencing costs declined, the pangenome approach was extended to eukaryotic genomes, including humans and *Arabidopsis thaliana* (Cao et al., 2011; Li et al., 2010; Sudmant et al., 2010). Over time, this concept evolved into a broader model encompassing the full genomic landscape of a population, species or clade. A somewhat unfortunate fact is that 'pangenome' these days more often refers to the collection of a specific set of genomes rather than the ideal of all reasonably common variants in a population.

## 2. The evolution of building pangenome graphs

One intuitive way to overcome the limitations of a linear reference genome is through genome graphs, which offer a compact data structure where sequences are represented as node-labelled graphs with edges connecting variants across multiple genomes. Unlike traditional linear representations, genome graphs preserve original coordinates by tracing paths through the sequence graph, accommodating both shared and unique genomic regions. The interpretability of pangenome graphs and their level of detail exist on a two-dimensional spectrum. At one extreme, highly abstract graphs (Figure 1a,b), such as those representing every nucleotide variant with numerous loops and alternative paths, may be difficult to understand and have limited practical use. At the other extreme, unaligned genomic sequences, while easy to interpret, can obscure meaningful genomic differences. Depending on the approach, fixed k-mer-based de Bruijn graphs (Figure 1a,b) and fully aligned multi-individual genomes (Figure 1c–h) represent different points along this continuum (Figure 1i).

### 2.1. Variation-first: Augmenting the reference

Early pangenome graph construction was constrained by practical realities: high-quality genome assemblies were expensive and rare, while catalogues of variation from resequencing projects were abundant. This imbalance led to reference-augmented approaches that embedded known variants into linear reference backbones. These methods typically integrate variant genotyping within the same workflow, though here we focus on the graph construction aspect.

Schneeberger et al. (2009) pioneered this concept with GenomeMapper, demonstrating that incorporating known polymorphisms could reduce mapping bias in *Arabidopsis thaliana*. The approach was later extended to the major histocompatibility complex region in humans (Dilthey et al., 2015), where high diversity makes linear references particularly inadequate. These early successes established the fundamental principle that graph representations could capture variation more faithfully than linear sequences. It was further generalised by several groups to the whole genomes of thousands of human individuals, with each group using distinct strategies (Eggertsson et al., 2017; Garrison et al., 2018; Rakocevic et al., 2019). GraphTyper (Eggertsson et al., 2017) iteratively realigned, clipped and unaligned short reads with an embedded genome graph for small variant calling. The VG toolkit (Variation Graph toolkit, (Garrison et al., 2018; Hickey et al., 2020) emerged as the first comprehensive open-source framework for this reference-augmented paradigm. VG constructs variation graphs by threading known variants from VCF files into reference genomes or directly from genome alignment, creating alternative paths that represent different allelic states. It supports complex structural variations, which include duplications and inversions, using a bidirectional cyclic graph. Graph Genome pipeline (Rakocevic et al., 2019) also supports structural variants (SVs) for genotyping with high speed, but it is limited to human genomes and is not openly distributed.

Due to the flexibility of constructing graphs from precomputed VCFs, VG has become the backbone of many pipelines. By incorporating variants derived from genome assemblies or filtered long-read alignments, VG-based workflows have been successfully applied across diverse species, including humans and crops (Liu et al., 2020; Qin et al., 2021; Sirén et al., 2021; Zhou et al., 2022).

### 2.2. The alignment-first paradigm: Towards unbiased representation

With decreasing costs and improving quality of long-read sequencing, generating multiple high-quality genome assemblies has become increasingly feasible, shifting the bottleneck from data generation to comparative analysis. This shift enabled a new paradigm in pangenome graph construction – moving from reference-based variant threading to graph building through whole-genome alignments directly, adopting an 'alignment-first' approach. Theoretically, this approach can reduce reference bias and better capture complex structural variations, including inversions, duplications and rearrangements that are hard to encode using VCF-based models.

Even before genome graphs were formally introduced, a multiple genome alignment (MGA) already served as an implicit representation of shared and divergent sequence features across assemblies. Multiple sequence alignments (MSAs) naturally lend themselves to representation as partially ordered sequence (POA) graphs (Lee et al., 2002), which have been extended into A-Bruijn

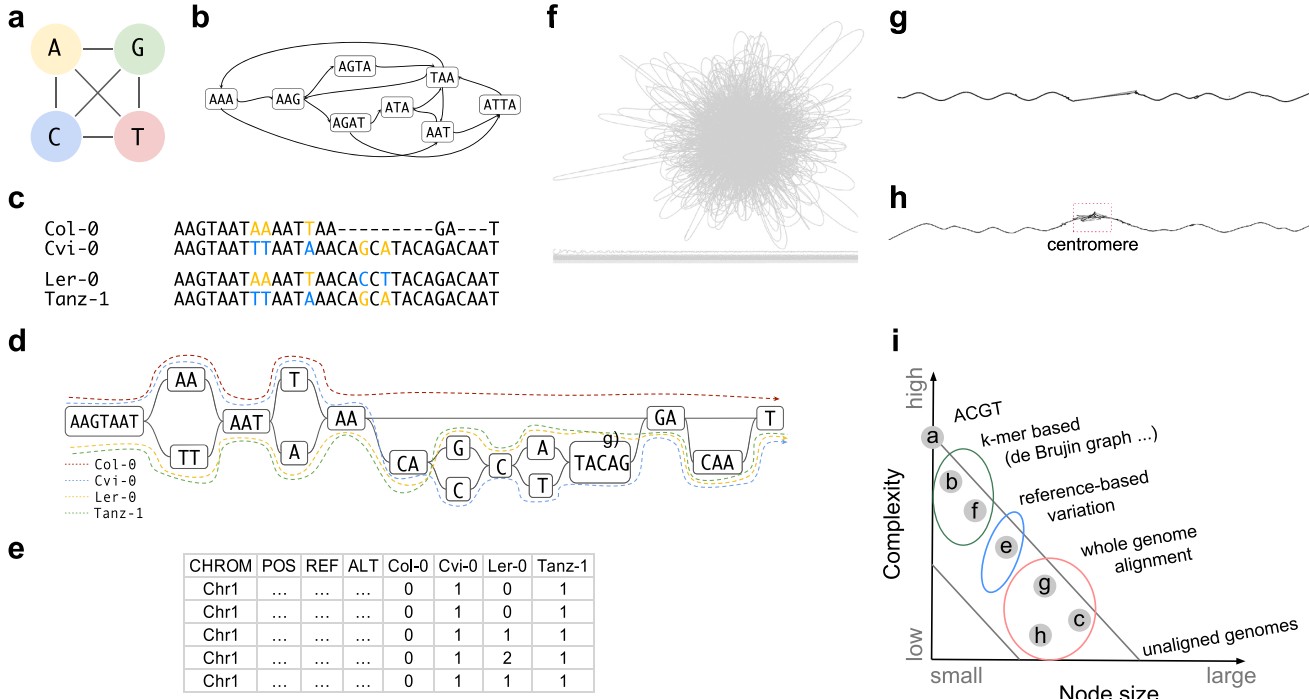

**Figure 1.** Multiple approaches to building pangenome graphs. (a) A graph that has only four nodes, corresponding to the four DNA bases, with all possible connections between the nodes. (b) A k-mer graph based on short sequences (here, triplets). (c) and (d) The same sequences are combined in different representations, which highlights the equivalency of a multiple genome alignment and a genome graph. (e) The same sequences shown in VCF format in a symbolic manner. (f), (g) and (h) represent the entire chromosome 1 from five *A. thaliana* individuals. (f) A Biforst 21-mer graph. (g) A Minigraph-Cactus graph. (h) A PGGB graph. (i) Summary of the trade-offs in node size and complexity for different types of graphs. Note that even with the same method, parameter choice will result in different graphs from identical sequences.

graphs (Raphael et al., 2004) and cactus graphs (Paten et al., 2011) to better accommodate genome rearrangements and duplications. Mauve (A. C. E. Darling et al., 2004) and TBA (Threaded Blockset Aligner) (Blanchette et al., 2004) represent some of the earliest efforts to align genome regions across multiple species. Vaughn and colleagues (Vaughn et al., 2022) recently used progressiveMauve (Darling et al., 2010) to align melon genomes and convert them into a genome graph for genotyping.

To bridge traditional alignment and graph construction, several intermediate tools have been developed. REVEAL (Recursive Exact-Matching Aligner) (Linthorst et al., 2015) employs a recursive exact-matching strategy to construct alignments, while tools like NovoGraph (Biederstedt et al., 2018) and Seqseq-pan (Jandrasits et al., 2018) utilise progressive or block-based alignment strategies to scale MGAs to a large number of genomes. ProgressiveCactus (Armstrong et al., 2020; Paten et al., 2011) dramatically improves scalability using a guide-tree-based alignment strategy. Its output can be used as alignment input to the VG toolkit, enabling the inclusion of large duplications and inversions in yeast (Garrison et al., 2018). This approach provided the first workflow for converting an MGA into a graph that can be used both to infer genotype information and to map short reads. SibeliaZ (Minkin & Medvedev, 2020) generalised these ideas based on information from a de Bruijn graph to construct improved MGAs.

The Human Pangenome Reference Consortium (HPRC) (Liao et al., 2023) has greatly advanced the field by releasing an initial pangenome draft from 47 humans, constructed using methods like Minigraph, Minigraph-Cactus and Pangenome Graph Builder (PGGB). Minigraph (Li et al., 2020) extended the minimap2 chaining algorithm to progressively add large SVs (>50 bp) into the graph. Minigraph-Cactus (Hickey et al., 2024) recruits the graph from Minigraph as a backbone. It then adds base-level alignments after clipping sequences that are highly divergent from a chosen reference sequence ('clipping' is the technical term for removing the portion of a query sequence that cannot be confidently aligned to the target genome). The details of these graphs will depend on the order of the input of sequences or the divergence between samples in the collection of genomes (Garrison & Guarracino, 2023), but it simplifies the graph structure and makes the graph suitable for downstream genotyping tasks. Similarly, ACMGA (AnchorWave-Cactus Multiple Genome Alignment) (Zhou et al., 2024) combines cactus with AnchorWave, which improves the alignment of long repetitive sequences in the plant genomes, for detection of large SVs (Song et al., 2022). Huijse et al. (2023) found that AnchorWave outperformed Minigraph-Cactus in producing alignments in the highly divergent MHC region of human genomes. The PGGB (Garrison et al., 2024) tries to capture all variations in the input sequences by constructing and all-to-all genome alignment by wfmash and rendering it with seqwish (Garrison & Guarracino, 2023) and GFAffix, then further consensus with smoothxg. While this approach offers a comprehensive representation of variations, the computational demands of all-to-all alignments are substantial. Instead of building a whole genome graph, PGR-TK (PanGenomic Research Took Kit) (Chin et al., 2023) rapidly constructs subgraphs of specific regions using data structures designed for long-read assembly (Chin & Khalak, 2019; Li, 2016); it was shown to be very fast in rebuilding the complex variations in MHC haplotypes, though its use demands substantial expertise for parameter tuning and result interpretation.

## 2.3. Scalable alternatives to whole-genome alignments

Over the past decade, the complexity and scalability challenges of constructing and querying large genome graphs have become increasingly apparent. As a result, researchers have explored pangenomes using analyses based on specific sequence blocks – such as orthologous genes or k-mers – rather than base-resolution DNA sequences. Different strategies have been developed to make pangenome analyses more scalable, each with its own trade-offs. K-mer based approaches are computationally efficient, making them attractive for large-scale comparisons. However, they sacrifice sequence context and struggle to distinguish between repeats, particularly in complex eukaryotic genomes. In contrast, gene-based methods are more interpretable and extensible across genomes but depend heavily on good gene annotation. Annotation quality in turn is dependent on a range of factors, such as the availability of RNA and proteomics data, whether a genome is from a taxon that contains other well-annotated genomes and so on. The good news is that ever more comprehensive sampling, at the level of individuals (tissues and conditions), populations, species and higher-order groupings will undoubtedly improve gene annotations.

In bacterial pangenomics, gene presence–absence matrices generated by orthogroup clustering with OrthoMCL have been the standard (Contreras-Moreira & Vinuesa, 2013; Li et al., 2003; Page et al., 2015). This strategy was subsequently extended by incorporating gene graphs in tools like PPanGGOLiN (Gautreau et al., 2020) and Panaroo with partitioned and fixed annotation error (Tonkin-Hill et al., 2020). Genome Complexity Browser visualised and quantified variability with orthogroup inference (Manolov et al., 2020). PanPA constructs graphs based on protein sequence alignments (Dabbaghie et al., 2023), and Pangene leverages rapid protein alignments to build gene graphs for eukaryotic genomes, enabling analysis of gene copy number changes and orientation – remarkably, it can build a graph from 100 human haplotypes in under one minute (Li et al., 2024).

Although implicit graphs constructed from fixed k-mers provide a valuable snapshot of genomic diversity, their resolution is inherently limited, and other tools have taken different routes. PanTools (Sheikhizadeh et al., 2016) detects homology groups with k-mers and builds a database for pan-proteome query, while PanKmer (Aylward et al., 2023) and Panagram (Benoit et al., 2025) decompose assembled genomes into a k-mer database with further ability to locate specific positions in assemblies. Furthermore, methods like Biforst (Holley & Melsted, 2020) and mdBG (Ekim et al., 2021) efficiently construct de Bruijn graphs for storage and rapid querying; they can be applied to genotype variable tandem repeats with short reads (Lu et al., 2021), though they fall short in accurately representing complete loci for downstream analyses (Andreace et al., 2023).

## 3. Variant calling in the graph era

Once a pangenome graph is constructed, it can serve as an enhanced reference for genotyping resequenced samples – either by aligning reads or matching k-mers – capturing a broader range of sequence variation than linear references. While many current tools rely on read mapping or k-mer comparison to identify SNPs and SVs, some have advanced to support haplotype reconstruction and the detection of novel variants – capabilities that are particularly effective with long-read resequencing.

Among these, one of the most widely adopted and versatile tools is the Variation Graph Toolkit (VG), which provides a comprehensive framework for mapping, small variant calling, and SV genotyping. VG has become popular since its first open-source release (Garrison et al., 2018; Hickey et al., 2020). It also reduces reference bias in ancient samples (Martiniano et al., 2020). Another VG module, Giraffe (Sirén et al., 2021), was developed as a successor of VG map to accelerate the process for large-scale genotyping. PHG (Practical Haplotype Graph) utilises established tools for mapping against linear references (e.g., GATK) for genotyping in the offspring of crops (Bradbury et al., 2022). DRAGEN (Dynamic Read Analysis for GENomics) (Behera et al., 2024) is currently the fastest for mapping and genotyping against pangenome references, exploiting hardware acceleration and tricks from machine learning, but it requires a commercial license. Apart from directly mapping to a graph, mapping reads to multiple references first and then injecting them into a graph based on mapping coordinates is another direction; one example is Gfa2bin (Vorbrugg et al., 2024) and cosigt (Bolognini et al., 2024), which uses node coverage across multiple references for genotyping with mapping by bwa (Li, 2013). Such approaches benefit from the maturity of linear reference mapping and the compatibility of their outputs with downstream graph-based analyses. Mapping long reads directly to genome graphs has become increasingly viable. Graphaligner (Rautiainen & Marschall, 2020) is the first tool to achieve long-read mapping to a graph with a seed-and-extend strategy, with much higher speed than VG. Minigraph (Li et al., 2020) can find approximate mapping locations without base-level alignment, while Minichain (Chandra et al., 2024) introduces a recombination penalty for long reads mapping to the graph.

To sidestep the computational cost of full mapping, many tools employ k-mer comparison strategies that match sequencing reads to known variants encoded in the graph. PanGenie (Ebler et al., 2022) and KAGE (Grytten et al., 2022, 2023) compare k-mers from reads to a pangenome graph to reduce run time and mapping bias. Ensemble Variant Genotyper (Du et al., 2024) is a framework designed to standardise the performance of various genotyping tools by accounting for the genomic features specific to plant species. Varigraph (Du et al., 2025) further optimised the k-mer-based approach with memory efficiency and extended the model for dosage estimation in autopolyploid genomes. A drawback is that these tools only genotype the known variations independently and thus cannot reconstruct the haplotypes in the population. To address this gap, Locityper (Prodanov et al., 2025) and cosigt (Bolognini et al., 2024) have been developed to utilise read alignment profiles to locate the closest haplotype in the graph.

Furthermore, SV calling directly from pangenome graphs remains a critical challenge. To overcome the issues, SVarp (Soylev et al., 2024) tackles this by locally assembling potential SV alleles from long-read data, while PALSS (Denti et al., 2025) augments the graph with the consensus from sample-specific long reads without mapping.

In summary, the field of pangenome graph construction is dynamic, with no single tool dominating; the optimal tool choice depends on the specific research objectives and the desired resolution. Reference-based variation graphs, for instance, facilitate population genetics analyses across extensive cohorts but may omit certain genomic variations. Tools like PGGB offer comprehensive graph representations; however, their complexity can pose challenges for downstream applications such as VG Giraffe alignment, necessitating tailored pruning strategies for effective read mapping. Notably, developing and benchmarking

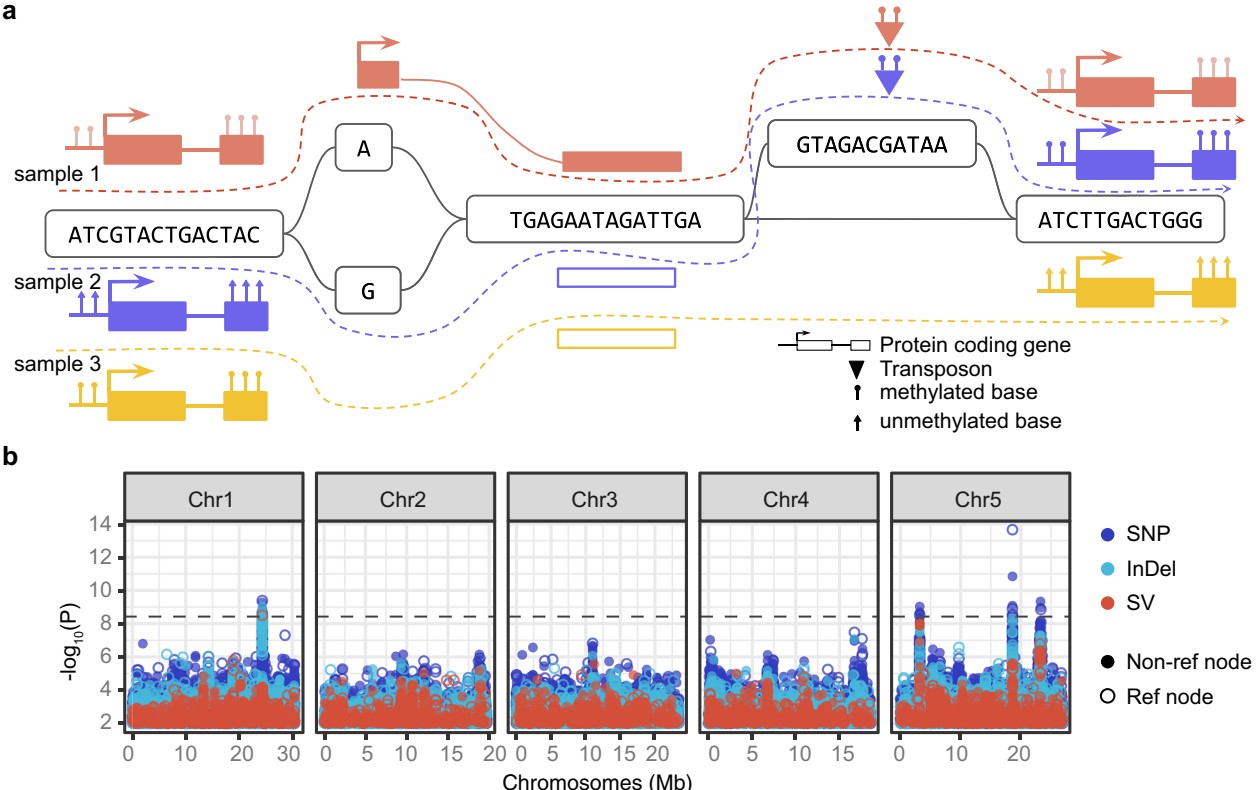

**Figure 2.** Functional pangenomics. (a) A pangenome graph can integrate diverse layers of functional annotations (e.g., genes, transposons, methylation level) in its reference coordinate system and serve as a unified platform for cross-genome comparison. (b) Graph nodes can be used directly for genome-wide association analyses. The colors represent different length-based categories for the node, while node shapes indicate whether the sequence originates from the chosen reference genome. Broken line indicates nominal significance threshold. This figure was adapted from (Vorbrugg, Bezrukov, Bao, Xian, et al., 2024).

efforts have predominantly centred on human genomics. Given that non-human species, including plant genomes, are often much more diverse than human genomes, there is a need for expanded evaluation across diverse species of tools for the building and use of pangenomes.

## 4. Functional pangenomics: Linking variation to mechanism

Reference bias not only affects variant discovery. Its shortcomings have knock-on effects in downstream functional analyses, including the comparison of chromatin accessibility, gene expression, or DNA methylation (Galli et al., 2025; Igolkina et al., 2025). Compared to the growing adoption of genome graphs for structural variation calling and genotyping, much more still needs to be done to take advantage of graph-based frameworks for functional genomics.

Grytten et al. (2019) implemented Graph Peak Caller to identify ChIP-seq peaks using a variation graph in *A. thaliana*, identifying more than twice as many base pairs absent from the linear reference than had been found with previous methods. DNA methylation studies have revealed analogous benefits – and also underscore the extent of reference bias in functional assays. In cattle, using the wrong reference genome can lead to substantial errors in methylation quantification, with up to ~2% global bias and large numbers of methylated cytosines being affected by breed-specific variation (MacPhillamy et al., 2024). In *Arabidopsis thaliana*, methylation profiling was even more sensitive to reference choice, with only ~88% of sites being consistent between reference

and focal strain, with one major reason being that transposable elements, which are prime targets of DNA methylation, have been much more active in this species than, for example, in humans (Igolkina et al., 2025). To address this, methylGrapher (Zhang et al., 2025) introduced the first graph-based approach for mapping bisulfite sequencing data. Compared to traditional methods such as Bismark, it uniquely identified 2.2–2.9 million $^{m}CpGs$ across five human samples, many of which were absent from the reference or misclassified as unmethylated before.

Reference bias also affects RNA-seq analysis. In *Arabidopsis thaliana*, expression estimates diverged for a subset of genes depending on whether reads were mapped to the reference genome or to the accession's own genome; these genes were strongly enriched for transposable elements and copy number–variable loci (Igolkina et al., 2025). Similar trends were observed in barley, but at an even higher rate, where mapping transcriptomic reads to a pan-transcriptome built with 20 genotypes improved the mapping rate by around 11% compared to a single linear reference (Guo, Schreiber et al. 2025). VG rpvg (Sibbesen et al., 2023) extends genome graph approaches to RNA-seq analysis by building spliced pangenome graphs and quantifying expression along haplotype-resolved paths (Sibbesen et al., 2023). These methods improve accuracy and enable haplotype-specific quantification, even without prior haplotype phasing, but they are ideally based on comprehensive pan-transcriptome annotation, which is absent in most species. Haplotype information in turn is immensely useful in outbred species, and perhaps even more so, in polyploid species with their complex allele ratios (Bao et al., 2022; Bird et al., 2025; Du et al., 2025).

Despite these advances, graph-based approaches to functional genomics remain in their infancy. Few tools have been developed, and most remain proof-of-concept applications limited to model species. Even where tools exist, broader adoption has been slow, partly due to the lack of comprehensive functional annotations and the complexity of graph-aware analytical workflows. Expanding these approaches across multiple omics layers – including methylation, expression, chromatin states and chromatin accessibility – and to diverse species with more complex genomes remains a critical challenge for future research (Figure 2).

## 5. Navigating the tangled graph: visualisation, comparison and scalability

Although there are multiple strategies for graph construction, most approaches now adopt the Graphical Fragment Assembly (GFA) format to store graph information. Unfortunately, querying large-scale pangenomes remains challenging due to the inherent complexity and enormous size of these graphs. For instance, the VG toolkit offers a versatile suite of functions to construct, convert and manipulate genome graphs, but even with VG, extracting information from Gb-scale pangenomes can be nontrivial. To overcome scalability issues, several specialised tools have been developed (Figure 3a). ODGI (Optimised Dynamic Genome/Graph Implementation) (Guarracino et al., 2022) implements scalable algorithms to visualise graphs at multiple resolutions, to extract specific loci and to compare path similarities. Meanwhile, tools such as Gretl (Vorbrugg et al., 2024) are designed to evaluate the quality of multiple graphs by providing a range of quantitative metrics for graph description and comparison. PANCAT (Dubois et al., 2025) characterises differences among variation graphs derived from the same sequence set using edit distance metrics.

On the visualisation side, early GUI-based tools like Bandage (Wick et al., 2015) and GfaViz (Gonnella et al., 2019) provide whole-graph views of assembly graphs but are limited when it comes to base level or Gb-scale pangenome graphs. VG view and VG viz can display sequences up to about 100 kb, whereas SequenceTubemap (Beyer et al., 2019) adopts an intuitive visualisation model (inspired by public transport network maps) to display variation graphs along with read mappings at the appropriate scale. Momi-G (Yokoyama et al., 2019) extends this concept for large-scale SV inspection in human variation graphs, and ODGI viz further expands on the VG viz layout by exporting rasterised images suitable for chromosome-scale genome graphs.

Efforts to integrate graph layouts with functional annotation are also emerging. For example, VRPG, a visualisation and interpretation framework for linear reference–projected pangenome graphs (Miao & Yue, 2025), extracts subgraphs based on reference path coordinates and annotations, while PPanG (Liu, Zhang et al., 2024) adapts the SequenceTubemap framework to display multiple genome annotations through embedded JBrowse2 components in real time. Additionally, Gfaestus (https://github.com/chfi/gfaestus) leverages GPU frameworks to visualise full graphs from projects like HPRC, and waragraph (https://github.com/chfi/waragraph) can integrate annotation information into ODGI layouts interactively.

Compared to graph construction, the visualisation and comparison of pangenome graphs have lagged significantly. While multiple tools exist for assembling and processing variation graphs, there is still no comprehensive, scalable and interactive visualisation framework that can handle large-scale pangenomes efficiently and connect the functional annotation (Figure 3b). As the pangenome expands to hundreds of individuals rapidly, it could even go beyond species, and extracting biological knowledge from the complex tangles in graphs requires better tools than what is currently available.

## 6. Conclusion and perspectives

The development of eukaryotic pangenomics has entered a transformative phase. Advances in sequencing technologies and assembly algorithms have made it feasible to generate high-quality genomes at population scale (Antipov et al., 2025; Cheng et al., 2024; Koren et al., 2024). As a result, pangenome references constructed from tens to hundreds of assemblies now exist for a growing number of species, including foundational species such as *Arabidopsis thaliana* (Kang et al., 2023; Lian et al., 2024; Wlodzimierz et al., 2023), key crops (Cheng et al., 2025; Guo et al., 2025; Hufford et al., 2021; Liu et al., 2020; Lynch et al., 2025; Zhou et al., 2022) and humans (Liao et al., 2023). Its application has shown that additional variations capture some of the heritability previously missed (Zhou et al., 2022), find more associations between variations and agronomic traits (Hufford et al., 2021) and can uncover the complex evolution history of well-studied loci (Bolognini et al., 2024).

Nevertheless, capturing the full spectrum of variation across a species in an unbiased and comprehensive way remains a challenge. While tools such as Minigraph-Cactus use iterative construction to simplify the process of graph alignment, they are sensitive to input order and tend to discard sequences that diverge too much from the reference – an issue especially problematic in high-diversity species (Cheng et al., 2025; Garrison & Guarracino, 2023). On the other hand, all-to-all alignment approaches, such as PGGB, provide more complete graphs but require substantial computational resources, making them impractical for datasets involving hundreds of genomes (Lynch et al., 2025). Similarly, genotyping tools face scalability constraints for large graphs: VG Giraffe, for instance, typically downsamples to 64 haplotypes prior to mapping (Sirén et al., 2021).

Progress in these areas depends on the availability of high-quality benchmark datasets for validation. Yet such resources are scarce in non-human species, and even in human genomics, benchmarking is often confined to a few well-characterised individuals (Dwarshuis et al., 2024). This creates systematic bias and limits our ability to assess how well genome graphs capture rare, complex or population-specific variation. Developing robust metrics and comparative frameworks to evaluate graph quality remains a crucial direction for the field.

Moreover, the continued reliance on biallelic SNP models restricts the development of population genetic theory capable of explaining the full complexity of pangenomic variation. However, SVs are generated by a wide range of distinct mutational mechanisms – including non-homologous end joining, non-allelic homologous recombination, template-switching, nested transposon insertions and tandem repeat expansion – that often result in multi-allelic loci rather than simple binary variants (Collins & Talkowski, 2025). In association studies, the assumption of biallelic variation also introduces confounding effects, particularly in the presence of genetic heterogeneity (Liu et al., 2025). Incorporating haplotype-aware models may reveal additional associations that are otherwise missed due to the underlying complexity of SVs and the impact of multi-allelic loci (Smith et al., 2025; Zhou et al., 2022). As such, there is a growing need to develop new population genetic models.

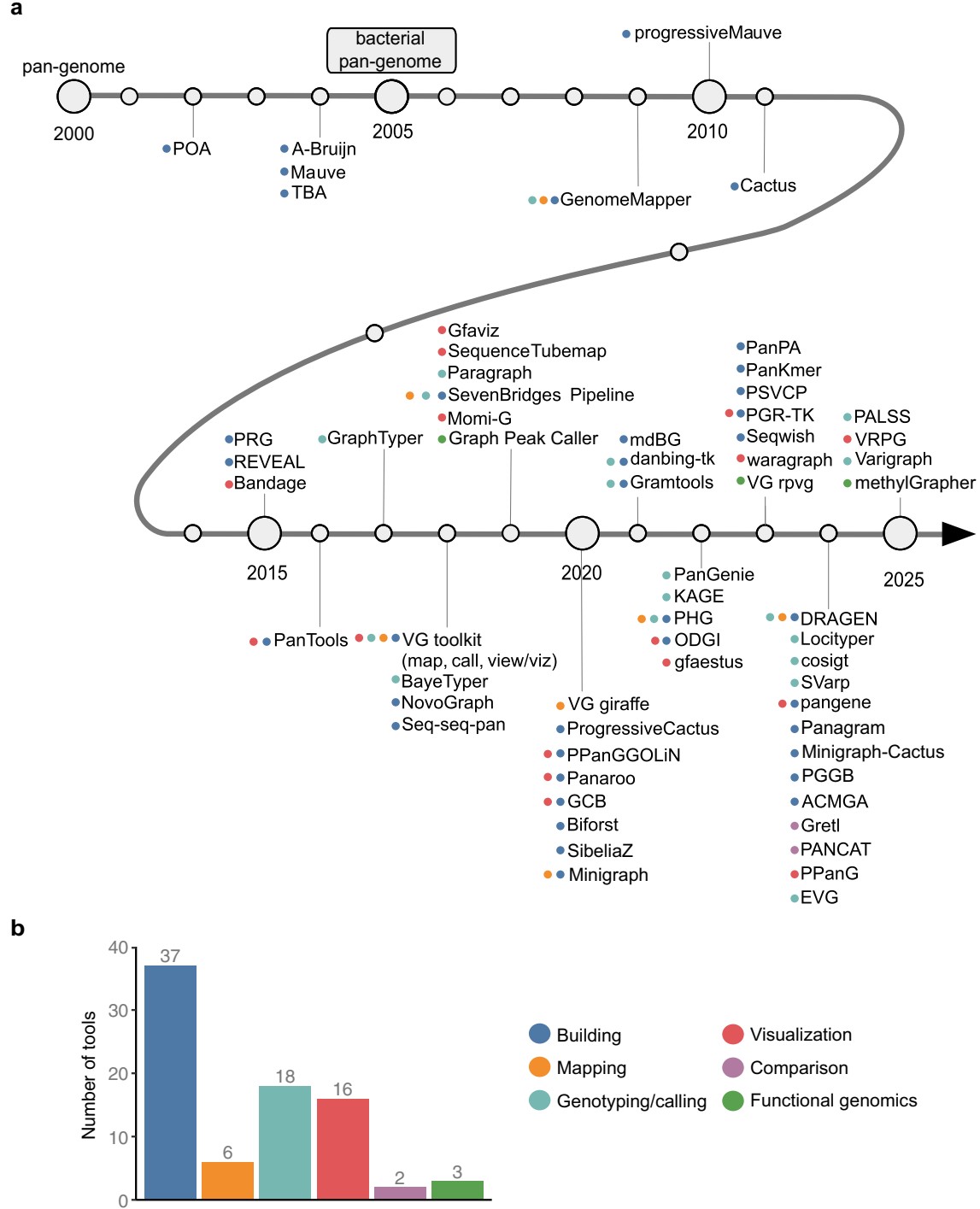

**Figure 3.** Timeline of pangenome graph algorithms. (a) The differently coloured circles indicate the main functions of tools; some workflows/tools may have multiple usages. For tools described in journals, we use the date of publication, but we note that many colleagues in this area are very generous and often release their tools long before formal publication. Given the rapid development in this area, it is perhaps not surprising that some tools only have a public GitHub repository. In the text, we provide hyperlinks as references. (b) The number of graph-based tools developed for different purposes.

Looking ahead, integrating pangenome graphs with evolutionary models remains a wide-open frontier. Ancient hybridisation, incomplete lineage sorting, and structural rearrangements complicate cross-species alignment, increasing the difficulty of graph construction. Yet global biodiversity sequencing projects (Lewin et al., 2018) are beginning to fill the tree of life with genome assemblies. Embedding phylogenetic history directly into graph construction – rather than treating it as a downstream layer – may help generate more meaningful, interpretable graphs.

In contrast to advances in variant calling, the interpretation, visualisation and benchmarking of pangenome graphs are still in early stages. There is still no equivalent of IGV for intuitive graph exploration, and widely adopted formats for complex variant representation are lacking. While tools like ODGI offer useful

summaries and visualisations, they lack interactivity and scalability for Gb-scale graphs. Even more critically, the integration of functional genomics with graph frameworks remains far behind. At present, RNA-seq, methylation, and chromatin accessibility data cannot be seamlessly analysed in graph-aware contexts. Bridging this gap will require unified, scalable methods for aligning and interpreting multi-omic data within graph-based references for genotype-phenotype association.

As we enter the next phase of pangenomic research, the field faces substantial computational and modelling hurdles. Yet the growing ecosystem of graph-based methods offers more than just an ever-expanding toolkit – it provides the foundation for a new paradigm in genomics. By embracing the full complexity of genomic variation, pangenome graphs have the potential to reshape how we conduct association studies, trace evolutionary history and interpret regulatory landscapes. Moving beyond reference bias and linear constraints, these graphs can unify population-scale diversity, functional readouts and comparative signals across the tree of life. Realizing this vision will require not only scalable tools and new theoretical frameworks but also sustained community efforts in benchmarking, visualisation and data integration. Still, the promise is profound: to build genomic models that reflect biological reality more accurately, and in doing so, to understand evolution in unprecedented detail.

**Open peer review.** To view the open peer review materials for this article, please visit http://doi.org/10.1017/qpb.2025.10028.

## Acknowledgements

We thank both our colleagues from the 1001 Genomes Project in the Weigel and Nordborg labs and the participants of the International Genome Graph Symposium (IGGSy'24) in Ascona 2024 for inspiration, and especially Andrea Guarracino and Erik Garrison from the University of Tennessee Health Science Center for extensive discussions.

**Competing interest.** D.W. holds equity in Computomics, which advises plant breeders. D.W. also consults for KWS SE, a globally active plant breeder and seed producer.

**Author contributions.** Z.B. wrote the draft of the manuscript. Z.B. and D.W. revised the manuscript.

**Funding statement.** Our work is supported by the International Max Planck Research School (IMPRS) 'From Molecules to Organisms' (Z.B.), the Novozymes Prize of the Novo Nordisk Foundation (D.W.), and the Max Planck Society.

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
