## [Reviewer Report]

Bao & Weigel present a well-written review of the development of analytical tools for pangenomes. It is a welcome overview of the different analytical tools, focusing on the rather technical, and complicated, aspects of analyses and interpretations of pangenomes.

1.The last sentence in the abstract refers to “reference-free evolutionary and population genomics”. I question the use of “reference-free”. What do the authors mean by this term? Will we ever be able to not rely on a reference in genomic analyses?

2.Introduction: In many parts of the introduction the authors use very long and complex sentences that can be hard to completely follow. I suggest that the author simplify the language of the introduction by simply converting the very long sentences into multiple shorter once. For example, the first sentence in the second paragraph of the introduction spans 5 lines and contains two “which sub-sentences”. If converted to 2-3 shorter sentences the line of thought in this sentence would be much easier to follow and would also be clearer.

3. First paragraph of the Introduction: What is “modern biology”, and is evolutionary research “modern”? Do the authors refer to the use of genomics in biology being “modern”?

4. Last sentence of the Introduction: Will you ever be able to get “all variants present in a species”? Is that even desired? When will a pangenome refer to something else than a “collection of a specific-set of genomes”? How would you overcome this?

5. On page 7 you refer to “annotation quality”. I do agree with the authors that this is a problem, but the authors fail to include that annotations in themselves are very problematic for a number of reasons and that many genome annotations rely on “well-annotated” model organisms, e.g. Arabidopsis. Thus, to overcome the problem of annotation quality a whole new sets of annotation tools and more comprehensive biological sampling would be needed. I think the authors can include a few sentences about this for clarity.

6. I miss a discussion (can be short) on the specific complexity with pangenomics in plants, such as how to deal with hybridizations and polyploidy. It might also be worth to shortly comment on the uneven availability of genomic resources for plants, e.g. the overrepresentations of crops.

7. Fig 1: I suggest that the authors change the letters of the different panels such that the panels are described in alphabetical order in the figure legend. Also, panel e lacks a description. Panel e is referred to last in the text so then why is it not just the last panel? The vcf format in panel d is not completely clear to me.

8. Fig 2 and 3: Please make such that they are referred to in the correct order. It now looks like Fig 3 is referred to before Fig 2. I also believe the figure legends have been swapped but that the figures are uploaded in the reverse order. Since I cannot see which figure is what in the proof I’m not 100% sure.

9. Fig 2 (which I think is Fig 3 in the text): The colors and shapes in panel 2 are difficult to distinguish. It is almost impossible to see the difference between the two blue and the filled shapes fill also the un-filled shapes. Please consider revising this figure.

10. Fig 3 (which I think is Fig 2 in the text): I cannot see any a or b panels. Why are some programs in bold? The lighter green and blue are difficult to distinguish in the timeline.

---

## [Reviewer Report]

This is an excellent and timely article. I thoroughly enjoyed reading it. The manuscript provides a clear and engaging progression through the development of pangenome graph methods, their applications, and associated challenges. It is very well written and well structured.

I do not have any major concerns. Below, I provide a set of minor editorial suggestions that could further improve clarity, readability, and consistency.

Page 3

• Sentence: “This insight led early on to the suggestion of producing synthetic reference genome sequences that represented all possible combinations of polymorphisms across the genome.” → Please provide a reference or an example. What does synthetic reference genome sequences refer to here?

Page 5

• Last paragraph, second sentence: “MSAs” should likely be “MGAs.”

• Please spell out “TBA” at first mention.

Page 6

• Please spell out “PGGB” and “ACMGA” at first use.

Page 7 and subsequent pages

• Please spell out “PGR-TK,” “DRAGEN,” “EVG,” “KAGE,” and “PALSS” at first mention.

References / Bibliography

• The following reference is duplicated:

Paten, B., Earl, D., Nguyen, N., Diekhans, M., Zerbino, D., & Haussler, D. (2011). Cactus: Algorithms for genome multiple sequence alignment. Genome Research, 21(9), 1512–1528.

Page 9

• First paragraph, third last sentence: “in humans ((Igolkina et al., 2024).” → remove extra parenthesis.

Figures

• Figure legends appear to be mismatched: the legend for Figure 2 corresponds to Figure 3, and vice versa.

---

## [Editor Report]

Thank you for submitting your manuscript to Quantitative Plant Biology. Your review was positively received by both reviewers, who both detail minor suggestions for improvement.

---

## [Reviewer Report]

Two very small things:

Fig1: Page 4, second to last line in the “The evolution of building pangenome graphs)” I think it should be Fig. 1c-h. Also Make sure that the letters in panel i are updated to reflect the new order of the panels.

Fig2. Double-check that panel b is in the format you want. I do understand your thoughts behind the original coloring and shapes in the figure. Thank you for clarifying that in your responses. I think the legend of the shapes and colors have not made it into the new version.

---

## [Editor Report]

Thanks for submitting this revised version to QPB. It is an interesting review which was well received.

Reviewer 1 has some very minor comments

Two very small things:

Fig1: Page 4, second to last line in the “The evolution of building pangenome graphs)” I think it should be Fig. 1c-h. Also Make sure that the letters in panel i are updated to reflect the new order of the panels.

Fig2. Double-check that panel b is in the format you want. I do understand your thoughts behind the original coloring and shapes in the figure. Thank you for clarifying that in your responses. I think the legend of the shapes and colors have not made it into the new version.